# Effects of Dietary Threonine Levels on Growth Performance, Biochemical Parameters, Muscle Quality, and Intestinal Microflora of Rice Field Eel (*Monopterus albus*)

**DOI:** 10.3390/ani15182643

**Published:** 2025-09-09

**Authors:** Denghang Yu, Yujia Liu, Jiaxiang Chen, Jincheng Wan, Jiaqi Zhang, Chi Zhang

**Affiliations:** Hubei Key Laboratory of Animal Nutrition and Feed Science, School of Animal Science and Nutritional Engineering, Wuhan Polytechnic University, Wuhan 430023, China; yudenghang1985@163.com (D.Y.); 18365970228@163.com (Y.L.); chenjiaxiang777@163.com (J.C.); m13329798888@163.com (J.W.); zhangjiaqi@whpu.edu.cn (J.Z.)

**Keywords:** *Monopterus albus*, threonine, growth performance, biochemical indicators, muscle quality, gut microbiota, correlation analysis

## Abstract

This study systematically evaluated the effects of dietary threonine levels on growth performance, muscle quality, digestive function, and gut microbiota in rice field eel. The results show that supplementation with 9 g/kg (0.9%) threonine can promote the growth of rice field eels, improve their ability to digest and absorb food, and enhance muscle quality and the composition of intestinal microbial flora. The study provides new insights into high-efficiency green nutritional additive development and offers practical guidelines.

## 1. Introduction

The rice field eel (*Monopterus albus*) is one of the important freshwater economic fish species. It has a high muscle-to-fat ratio, with a protein content of up to 18–20% and a fat content of less than 1%. This combination of high nutritional value and delicate taste contributes to its popularity among consumers. The rice field eel has formed a complete industrial chain in Hubei, Hunan, Jiangxi, and Anhui provinces. Consequently, it has become one of the leading freshwater economic aquaculture species in China [1]. Muscle constitutes the primary edible portion of the rice field eel, with its quality determined by both nutritional content and sensory attributes [2]. Numerous studies have demonstrated that these traits are dynamic and closely correlate with the composition of feed ingredients, particularly the levels of protein and fat sources [3,4]. Rice field eels are carnivorous fish that require over 40% protein in their feed [5]. They require high-quality protein to support optimal growth and development. Proteins are made up of amino acids; therefore, the protein requirement of rice field eels is fundamentally a requirement for essential amino acids (EAAs) [6]. Among these, lysine, methionine, and threonine are commonly identified as the first limiting amino acids. A deficiency or excess of these amino acids can disrupt the amino acid balance, leading to decreased protein utilization, growth retardation, abnormal fat deposition in muscle, and potentially induce oxidative stress and immunosuppression [7]. Therefore, achieving the optimal protein profile by supplementing essential amino acids has emerged as a critical strategy for improving the muscle quality of rice field eels, reducing the feed conversion ratio, and promoting sustainable aquaculture practices.

Threonine (Thr), one of the essential amino acids, is unique in that it does not require deamination or transamination. Thr plays a crucial role in the normal growth and metabolism of animals. Animals are unable to synthesize Thr and must obtain it through dietary supplementation [8]. Recent studies on aquatic animals have demonstrated that Thr not only promotes fish growth but also enhances muscle quality and modifies the composition of intestinal flora. Numerous studies indicate that the exogenous addition of Thr can increase the weight gain rate of largemouth bass (*Micropterus salmoides*) [9] and Indian catfish (*Heteropneustes fossilis*) [10], as well as the specific growth rate of intermediate-stage grass carp (*Ctenopharyngodon idella*) [11] and large yellow croaker (*Pseudosciaena crocea*) [8]. To enhance muscle quality, the composition and concentration of threonine are essential indicators for evaluating the nutritional value of meat. Supplementing feed with Thr can enhance the textural characteristics of Jian carp (*Cyprinus carpio* var. *jian* ) muscle [12] and upregulate genes associated with myogenesis in hybrid catfish (*Silurus asotus*) [13]. Incorporating an appropriate level of Thr into the feed can enhance the body and amino acid composition of grass carp [14]. Additionally, research on Nile tilapia (*Oreochromis niloticus*) indicates that dietary Thr can enhance the flavor of fish flesh [15]. In regard to intestinal health, Thr is predominantly found in the small intestine of mammals, indicating its role in intestinal function. Studies have shown that an appropriate level of Thr can elevate the activities of lipase, amylase, trypsin, and chymotrypsin in blunt snout bream (*Megalobrama amblycephala*) [16]. Furthermore, threonine can rebalance the gut microbiota in animals and positively impact microbial health. Supplementing feed with threonine can increase the relative abundance of *Lactobacillus* in the animal gut [17]. Under stress conditions, the addition of 0.3% Thr to low-protein feed can restore gut microbial diversity and significantly enhance the abundance of potentially beneficial microorganisms [18]. Therefore, when Thr is used as a feed additive for aquatic animals, it can significantly enhance their overall health.

Currently, the Thr requirements for various aquatic species have been established. A deficiency of dietary threonine can suppress growth performance by reducing animal appetite, resulting in slow growth, low feed utilization, and even damage to digestive organs [19]. On the other hand, excess dietary Thr can lead to the excessive accumulation of Thr and its metabolites in fish, accelerating enzymatic reactions and causing further amino acid accumulation, thereby inducing toxic effects [20]. For instance, the optimal Thr requirement for grass carp is 1.37% [14], while juvenile Nile tilapia require 1.33% [15], and blunt-snout bream require 1.57% [16]. Consequently, establishing the optimal Thr requirement in this study is essential. Currently, no reports exist regarding the effects of Thr on the muscle quality and intestinal microbiota of rice field eels. Thus, this study utilized artificially bred rice field eels (18.47 ± 0.11 g) as research subjects to investigate the effects of dietary Thr levels on growth performance, biochemical indices, and muscle quality. A 16S rRNA gene sequencing technology was employed to compare the structure and composition of intestinal microbial communities in rice field eels subjected to varying Thr levels. The optimal threonine requirement for rice field eels was established, providing a reference for improving farming strategies and enhancing muscle quality.

## 2. Materials and Methods

### 2.1. Experimental Design and Feed Preparation

To ensure the accuracy of experimental results and avoid physiological data deviations caused by gender changes, all-male eels were selected as the research subjects. Rice field eels were determined to be male by body differences and the protrusion of genital pores. Additionally, to ensure all test eels were male, the condition of gonads is observed during the sampling process. Healthy male adult rice field eels, with an initial body weight of 18.47 ± 0.11 g, were randomly assigned to six groups: a control group (T1) that contained no threonine, and five experimental groups (T2, T3, T4, T5, and T6) that received increasing levels of threonine supplementation. Each group, comprising three replicates of 30 eels housed in length 1.5 × 2 m net cages, was subjected to a trial lasting 60 days. Feed composition and nutrient levels are presented in Table 1. The basal diet included fish meal, peanut meal, corn protein powder (protein), wheat flour (carbohydrate), and a mix of fish oil and soybean lecithin (lipid). Mixed amino acids were added based on the whole-body amino acid profile of rice field eels for reference. L-Threonine was supplemented to the basal diet at levels of 0.3%, 0.6%, 0.9%, 1.2%, and 1.5% (3 g/kg, 6 g/kg, 9 g/kg, 12 g/kg and 15 g/kg), balanced with glycine, to prepare six isonitrogenous and isoenergetic diets. The added dose is based on He Zhigang’s research [8]. The measured content of Thr in the basic feed (T1) was 0.855%. None of the premixes contained Thr. All ingredients were obtained from Wuhan CP Aquaculture Co., Ltd. (Wuhan, China) They were ground, sifted through a 40-mesh sieve, weighed, blended in a stepwise manner, then pelletized into 2.0 mm pellets, air-dried, labeled, and stored at −5 °C.

### 2.2. Aquaculture Management

The rice field eels were obtained from Hubei Xiantao Rice Field Eel Industry Group Co., Ltd. located in Xiantao, China, and were acclimatized for a week at the Xiantao Rice Field Eel Industry Technology Research Institute. Water hyacinth, provided by the same company, can purify water quality, regulate temperature, and provide hiding spaces [21]. Pest control for the water hyacinth was carried out as needed. The water hyacinths infested by pests were directly removed from the water body using manual dredging. The culture water was sourced from filtered local wells and maintained under controlled parameters: temperature 25–30 °C, pH 6.50–7.50, ammonia ≤ 0.15 mg/L, nitrite ≤ 0.05 mg/L, and dissolved oxygen > 6 mg/L, all of which meet Chinese fishery standards. The feed ration was set at 5% of the rice field eel’s body weight. During the initial farming phase, approximately 27 g of feed was supplied per tank, which was later increased to about 52.5 g. FCR was calculated by subtracting the amount of uneaten feed from the total quantity provided. Fish were fed once daily (18:00). Uneaten feed was removed at 06:00 the following day.

### 2.3. Sample Collection and Determination of Biochemical Indices

After a 24-h fasting period, rice field eels were anesthetized using 30 mg/L MS-222 (Tricaine methanesulfonate, M14788, sourced from AbMole in Shanghai, China) prior to sampling. All eels in each culture tank (70 cm × 35 cm) with a volume of 90 L were counted and individually weighed to determine final body weight (FW), weight gain ratio (WGR), specific growth rate (SGR), feed conversion ratio (FCR), and protein efficiency ratio (PER). Five eels per tank were randomly selected for measurements of body weight and length. Following tail amputation for blood collection, the eels were dissected on ice, with visceral and liver weights recorded for morphometric analysis. Tissue samples from the muscle, liver, and intestine were rapidly frozen in liquid nitrogen and stored at −80 °C; then, tissues were used for the determination of catalase (CAT, A007-1-1), superoxide dismutase (SOD, A001-3-2), malondialdehyde (MDA, A003-1-2), total antioxidant capacity (T-AOC, A015-2-1), trypsin (A080-2-2), amylase (AMS, C016-1-2), and lipase (LPS, A054-2-1) activities. Blood samples were kept at 4 °C for 24 h; serum was subsequently separated and stored at −80 °C for analysis of albumin (ALB, A028-2-1), total protein (TP, A045-2-2), glutamic oxaloacetic transaminase (GOT, C010-2-1), and glutamic pyruvic transaminase (GPT, C009-2-1). All commercial kits were from Nanjing Jiancheng Bioengineering Institute (Nanjing, China).

### 2.4. Calculation of Growth Performance and Morphological Parameters

Specific growth rate (SGR, %/d) = (LnW_t_ − LnW_0_)/t × 100,(1)

Feed conversion ratio (FCR) = F/(W_t_ − W_0_),(2)

Viscerasomatic index (VSI, %) = (VW/W) × 100,(3)

Hepatosomatic index (HSI, %) = (HW/W) × 100,(4)

Condition factor (CF, g/cm^3^) = W/L^3^,(5)

Weight gain ratio (WGR, %) = (W_t_ − W_0_)/W_0_ × 100, (6)

Protein effciency ratio (PER, %) = (W_t_ − W_0_)/(F × CP) × 100(7)

In these formulas, W_0_ and W_t_ represent the initial and final average weights (g) of the eels; t denotes the culture duration (days); F refers to the average feed intake (g); CP indicates the crude protein level (%); VW, HW, and W correspond to visceral weight, liver weight, and whole-body weight (g), respectively; and L stands for body length (cm).

### 2.5. Determination of Nutritional Components

The nutritional composition of feed and fish samples were analyzed following standard methods: the determination of moisture is carried out according to GB/T 6435-2014 using the drying method [22], the determination of crude protein is carried out according to GB/T 6432-2018 using the Kjeldahl method [23], the determination of crude fat is carried out according to GB/T 6433-2006 using the Soxhlet extraction method [24], and the determination of crude ash content is carried out according to GB/T 6438-2007 using the ashing method [25].

### 2.6. Analysis of Muscle Texture Parameters

Three rice field eels were selected from each culture tank, and dorsal muscle samples measuring 0.5 cm × 0.5 cm × 0.5 cm were collected. A texture analyzer (TMS-PRO, FTC, Sterling, VA, USA) was utilized to assess various textural parameters, including hardness, gumminess, chewiness, adhesiveness, cohesiveness, and springiness. The testing speed was established at 20 mm/min, with a contact force of 0.1 N and a deformation rate of 50%.

### 2.7. Measurement of Related Muscle Fiber Growth Gene Expression Levels

The rice field eel genome sequence information (NCBI database) was utilized to select RPL17 as the reference gene. Primers were designed using Primer 5.0 software and synthesized by Qingke Biotechnology Co., Ltd. (Beijing, China), with corresponding gene sequences presented in Table 2. Total RNA extraction was performed using the RNAiso Easy kit (9109). Complementary DNA (cDNA) was synthesized via reverse transcription following the PrimeScript RT Reagent Kit with gDNA Eraser (Perfect Real Time) protocol (RR047A). Real-time quantitative PCR (qPCR) was conducted on a Roche LightCycler 480 system with the TB Green^®^ Premix Ex Taq II kit (RR820A), adhering strictly to manufacturer instructions. Thermal cycling conditions comprised 95 °C for 30 s (initial denaturation), followed by 40 cycles of 95 °C for 5 s and 60 °C for 30 s. Relative gene expression levels were calculated using the 2^−ΔΔCt^ method [26]. All biochemical kits were supplied by TaKaRa Biotechnology Co., Ltd. (Shanghai, China).

### 2.8. Analysis of Intestinal Tissue Structure

Intestinal tissues were fixed in 4% paraformaldehyde for a minimum of 48 h prior to being transported to Servicebio Company (Shanghai, China) for tissue sectioning. Subsequent processing involved dehydration through a graded ethanol series and clearing in xylene. Samples were then embedded in paraffin and sectioned at 4 μm thickness using a microtome. Sections were stained with hematoxylin and eosin (H&E) and evaluated under an Olympus BX53 optical microscope (Tokyo, Japan). Morphometric measurements of villus width (VW), villus height (VH), and muscle layer thickness (MT) were conducted using ImageJ software V1.8.0.112 (2019) [14].

### 2.9. Analysis of Intestinal Microbiota

Twenty-four hours post-feeding, six individuals per replicate tank across all groups were randomly selected for euthanasia and dissection. Posterior intestinal luminal contents were collected aseptically into sterile 2 mL cryovials and immediately snap-frozen in liquid nitrogen prior to storage at −80 °C for microbiological analysis. Sequencing of the 16S rRNA gene was performed by Shanghai Majorbio Bio-pharm Technology Co., Ltd. (Shanghai, China). The sequencing protocol involved: (1) sample thawing from −80 °C, (2) homogenization of a 50 mg aliquot, (3) nucleic acid extraction using 725 μL SLX-Mlus buffer (PD090) from Shanghai Cinotech Biotech Co., Ltd. (Shanghai, China) with centrifugation, and (4) purification of total intestinal DNA from the resulting supernatant. High-throughput sequencing was conducted on an Illumina MiSeq PE300 platform (San Diego, CA, USA), with resulting sequence data deposited into the NCBI Short Read Archive (SRA) database.

### 2.10. Data Processing and Analysis

Data are presented as mean ± standard deviation (SD; n = 3). One-way analysis of variance (ANOVA) was conducted using SPSS version 26.0 (SPSS Inc., Chicago, IL, USA), followed by Duncan’s multiple range test for post-hoc comparisons. Statistical significance was set at *p* < 0.05.

## 3. Results

### 3.1. Growth Performance

As shown in Table 3, with the increase of dietary threonine levels, the FW, WGR, and SGR initially increased and then decreased (*p* < 0.05), reaching the maximum in the T4 group (*p* < 0.05). The FCR initially decreased and then increased with the rise of dietary threonine levels, and the T4 group was significantly lower than the control group (*p* < 0.05). As threonine levels increased, the PER initially increased and then decreased; the T4 group was significantly higher than the other groups (*p* < 0.05). As shown in Figure 1, utilizing WGR, SGR, FCR, and PER as evaluation indicators, the broken-line regression analysis revealed that the optimal dietary threonine requirement for rice field eel was 0.75–0.87%.

### 3.2. Morphological Indices and Whole-Body Composition

As shown in Table 4, the HSI and VSI of the rice field eel exhibited a decreasing trend with increasing dietary threonine levels; the T4 group was significantly lower than the control group (*p* < 0.05). As indicated in Table 5, the crude protein content initially decreased and then increased with the rise of threonine levels; the T4 group was significantly higher than the control group (*p* < 0.05). The crude fat content initially increased and then decreased with the rise of threonine levels; the T4 group was significantly lower than the control group (*p* < 0.05). The crude ash and moisture contents in the T6 group were significantly higher than those in the control group (*p* < 0.05).

### 3.3. Serum and Hepatic Biochemical Indices

As indicated in Table 6, with the increase in dietary threonine levels, the serum TP and ALB contents significantly increased (*p* < 0.05). With the increase in dietary threonine levels, the serum GPT activity of rice field eels initially decreased and then increased; the T4 group was significantly lower than the T5 group (*p* < 0.05). With the increase in dietary threonine levels, the GOT activity in the liver initially increased, then decreased, and then increased again; the T4 group was significantly higher than other groups (*p* < 0.05). As threonine levels increased, the GPT activity in the liver initially rose and then declined; the T4 group was significantly higher than the control group (*p* < 0.05).

### 3.4. Antioxidant Indices in the Liver, Intestine, and Muscle

As shown in Figure 2, with increasing threonine levels, the hepatic CAT and SOD activities in the T4 group significantly increased (*p* < 0.05). The hepatic MDA content decreased with rising threonine levels, and the MDA levels in the T4 and T5 groups were significantly lower than those in the control group (*p* < 0.05). No significant differences were found in hepatic T-AOC among the groups (*p* > 0.05).

As shown in Figure 3, with increasing threonine levels, the CAT activity and T-AOC initially increased and then decreased; the CAT and SOD activities, along with the T-AOC capacity in the T4 group, were significantly higher than those in the control group (*p* < 0.05). The MDA content initially decreased and then increased with rising threonine levels, and the MDA content in the T4 and T5 groups was significantly lower than that in the control group (*p* < 0.05).

As shown in Figure 4, the CAT activity in the muscle initially increased, followed by a decrease with rising threonine levels; the T4 group exhibited significantly higher CAT activity than the control group (*p* < 0.05). No significant differences were found in MDA content, SOD activity, or T-AOC capacity among the groups (*p* > 0.05).

### 3.5. Analysis of Muscle Texture Parameters

As shown in Figure 5, with increasing threonine levels, the intramuscular cohesiveness, springiness, gumminess, chewiness, and adhesiveness of the T4 and T5 groups were significantly improved (*p* < 0.05), whereas the muscle hardness of the T3 group was significantly reduced (*p* < 0.05).

### 3.6. Expression Levels of Genes Related to Myofiber Growth

As shown in Figure 6, with increasing threonine levels, the relative expression levels of the MyoG, MyoD1, and MYF5 genes initially rose and then declined, peaking in the T4 group (*p* < 0.05). The relative expression level of the MRF4 gene displayed an upward trend with increasing threonine levels, significantly peaking in the T6 group (*p* < 0.05).

### 3.7. Intestinal Digestive Enzyme Activities

As shown in Figure 7, dietary threonine levels had no significant effect on the amylase activity in the intestine (*p* > 0.05). With increasing threonine levels, the lipase and trypsin activities in the intestine initially increased and then decreased. The lipase and trypsin activities in the T4 group were significantly higher than those in the other groups (*p* < 0.05).

### 3.8. Intestinal Histomorphology

Figure 8 illustrated the morphological changes in the transverse section of rice field eel midgut in response to dietary threonine levels. In groups T1, T2, and T3, the intestinal villi exhibited shedding, becoming fewer and shorter. Group T4 exhibited intact cellular structures of intestinal villi, which were slender, finger-shaped, and displayed favorable morphology. Group T6 demonstrated a reduction in the number of intestinal villi, accompanied by partial shedding.

Based on the intestinal histomorphological indices presented in Figure 9, as the threonine level increased, the villus height (VH) initially rose and then declined. The villus width (VW) in group T4 was significantly higher than that in the control group (*p* < 0.05). The intestinal muscularis thickness (MT) in groups T5 and T6 was significantly lower than that in the control group (*p* < 0.05).

### 3.9. Intestinal Microbial Diversity

Based on a comprehensive analysis of growth performance, muscle quality, and digestive enzyme activity, the T4 group demonstrated superior growth and higher digestive enzyme activity compared to both the control and T5 groups. Consequently, intestinal microbiota analyses were conducted on rice field eel from the T1, T4, and T5 groups. Figure 10A illustrated that all samples (T1, T4, T5) collectively contained 332 operational taxonomic units (OTUs), with 97 shared among them. The T4 group exhibited the highest OTU count, and the degree of OTU overlap between samples reflects the similarity of microbial communities. Notably, the T4 group also possessed the highest number of unique OTUs, suggesting a more diverse array of microbial species. Alpha diversity index analyses (Figure 10B–E) revealed no significant impact of dietary threonine levels on the ACE or Chao indices (*p* > 0.05). However, the Shannon index was significantly higher in the T4 group than in the T5 group (*p* < 0.05). Principal coordinate analysis (PCoA) based on BrayCurtis dissimilarity (Figure 10F) showed distinct clustering patterns of gut microbiota among the T1, T4, and T5 groups, demonstrating significant differences in microbial community composition among these groups.

### 3.10. Composition and Differential Analysis of Gut Microbiota

As illustrated in Figure 11A, at the phylum level, *Proteobacteria*, *Fusobacteria*, *Firmicutes*, and *Bacteroidetes* consistently ranked as the top four dominant phyla across the T1, T4, and T5 groups. Figure 11B illustrates that at the genus level, *Pelomonas*, *Acinetobacter*, *Pseudomonas*, and *Cetobacterium* predominated as the four most abundant genera in all three experimental groups. The heatmap analysis of the top 21 genera with high relative abundance in the gut microbiota of rice field eel (Figure 11C) revealed distinct clustering patterns among the T1, T4, and T5 groups, suggesting significant intergroup variations in microbial composition (*p* < 0.05). As shown in Figure 11D, LEfSe analysis identified six significantly different bacterial genera (*p* < 0.05) with LDA scores exceeding 2.0, including *Clostridium sensu stricto 13*, *Brevinema*, *Sphingobacterium*, *Methylocella*, *Corynebacterium*, and *Dietzia*.

### 3.11. Correlation Analysis

Pearson correlation analysis was performed on the obtained data, with the results presented in Figure 12. Analysis of intestinal differential bacterial genera and growth performance in rice field eel indicated that *Corynebacterium* and *Methylocella* exhibited positive correlations with both WGR and SGR, whereas *Brevinema* demonstrated a significant negative correlation with protein efficiency (*p* < 0.05). Regarding muscle texture characteristics, *Corynebacterium* and *Methylocella* displayed positive correlations with hardness, elasticity, cohesiveness, gumminess, adhesiveness, and chewiness. In terms of muscle growth gene expression, *Corynebacterium* exhibited a significant positive correlation with MRF4 (*p* < 0.05), while *Clostridium sensu stricto 13*, Brevinema, and Dietzia were significantly negatively correlated with MRF4, MyoD1, and MyoG expression (*p* < 0.05).

### 3.12. KEGG Functional Prediction Analysis of Gut Microbiota

Functional annotation of intestinal microbiota in rice field eel was carried out using KEGG analysis. Among KEGG Level 1 pathways (Figure 13A), metabolic pathways exhibited the highest average relative abundance, followed by environmental information processing. For the Level 2 pathways (Figure 13B), global and overview maps, along with carbohydrate metabolism, predominated in relative abundance, with amino acid metabolism ranking second. The investigation specifically targeted three metabolic pathways illustrated in Figure 13C: glycine, serine, and threonine metabolism; pyruvate metabolism; and glyoxylate and dicarboxylate metabolism. Figure 13D suggests that *Corynebacterium* and *Methylocella* stimulate the core metabolic cascade of glycine/serine/threonine, pyruvate, and glyoxylate by upregulating the activities of threonine aldolase, glyoxylate reductase, and isocitrate lyase. This metabolic activation supplies essential energy, amino acids, and one-carbon units to support muscle development in rice field eel, consequently enhancing both growth performance and meat quality. *Clostridium sensu stricto 13*, *Brevinema*, *Dietzia*, and *Sphingobacterium* suppress the aforementioned pathways, impairing carbon/nitrogen substrate utilization efficiency and downregulating the expression of MyoD1, MyoG, and MRF4, resulting in reduced protein efficiency and compromised muscle quality in rice field eel.

## 4. Discussion

### 4.1. Comparing the Effects of Different Levels of Threonine on Growth Performance

The rice field eel is widely prized for its delicate texture and palatable flavor. In intensive aquaculture systems, the decline in meat quality due to high stocking densities diminishes consumer preference. Consequently, optimizing growth performance and enhancing muscle quality in rice field eels is essential. The WGR and FCR are critical metrics for assessing growth efficiency. Experimental findings revealed that threonine-deficient diets significantly impaired the growth of rice field eels. Progressively increasing dietary threonine levels substantially improved FW, WGR, and SGR, while simultaneously reducing the feed conversion ratio. Peak growth performance was observed in the T4 (0.9%) group, which is consistent with prior studies on Nile tilapia (*O. niloticus*) [27] and European sea bass (*Dicentrarchus labrax*) [28]. These studies demonstrate that dietary threonine deficiency inhibits protein synthesis, impairs normal growth and development, and may reduce antioxidant production, thereby exacerbating growth retardation. When the threonine content in feed reaches or exceeds 1.2%, the WGR, SGR, and PER of rice field eel show a declining trend, which is consistent with observations in grass carp (*C. idella*) [14] and Wuchang bream (*M. amblycephala*) [16]. This phenomenon may be due to excessive threonine or its metabolites in the fish’s amino acid pool, which could accelerate enzymatic reactions, leading to toxic amino acid accumulation. Furthermore, because ammonia is biologically toxic, its elimination requires additional energy expenditure, consequently reducing the SGR [29]. Studies have also shown that the amino acid balance in feed significantly influences amino acid absorption efficiency in organisms. Excessive threonine intake may disrupt this balance, impair nutrient absorption, and consequently lead to reduced growth rates [30]. Through broken-line regression analysis using WGR, SGR, FCR, and PER as evaluation metrics, the optimal dietary threonine requirement for rice field eel was determined to be 0.75–0.87%. This value is notably close to the 0.8% requirement [31] of the red drum in the United States (*Sciaenops ocellatus*) and the 1.03% requirement of the striped bass (*Lateolabrax maculatus*) [32].

The VSI and CF, which serve as key indicators for evaluating energy intake in fish, are also extensively used to assess the quality of aquatic products. This study revealed that the VSI of rice field eel declined with increasing dietary threonine levels, with the T4 group showing significantly lower values compared to the control group. No statistically significant differences in the condition factor were observed across all groups. These findings contrast with previous observations in grass carp [14]. The observed variations in threonine requirements across studies may arise from multiple factors, including fish species, size differences, basal diet composition, crystalline amino acid supplementation, experimental protocols, and aquaculture conditions.

### 4.2. Analysis of Whole-Body Composition and Muscle Quality

Muscle constitutes the primary edible portion of aquatic species and is directly correlated with economic value. Fish muscle primarily comprises proteins, lipids, moisture, and various other components, with their concentrations reflecting the nutritional quality of the tissue and serving as critical indicators of muscle development [2,33]. Our findings demonstrate that the crude protein content of rice field eel exhibited an initial increase followed by a decline as threonine levels rose, reaching optimal values in the T4 group. This implies that appropriate threonine supplementation enhances growth performance by improving nutrient retention. Comparable outcomes have been reported in studies on GIFT tilapia [34] and Jian carp (*C. carpio* var. *jian*) [35]. The mechanism may involve threonine supplementation under deficient conditions that optimizes dietary amino acid balance, thus improving feed utilization efficiency and stimulating protein synthesis. Conversely, excessive threonine results in the oxidative catabolism of surplus amino acids for energy production, generating excretory urea and ammonia. The crude fat content in rice field eel initially increased and then decreased as threonine levels rose, with the T4 group exhibiting significantly lower values than the control. This trend diverges from observations in grass carp, likely because the threonine intake in the T4 group optimally satisfied the amino acid balance–gluconeogenesis requirement, thereby directing excess carbon skeletons into the TCA cycle for energy oxidation and consequently suppressing lipid synthesis. The T6 group exhibited significantly higher crude ash and moisture content compared to the control, aligning with the phenomenon of moisture–ash co-elevation reported in Jian carp [35] under conditions of high methionine. This may result from excessive threonine stimulating urea cycle activity, leading to elevated ammonia nitrogen and organic acid levels that increase tissue osmotic pressure, promote water retention, and enhance mineral deposition—ultimately resulting in concurrent increases in both crude ash and moisture levels.

Fish muscle is not only nutritionally valuable, but its textural properties also serve as crucial indicators for assessing meat quality. These properties measure the forces required to compress and shear the muscle, characterizing attributes such as hardness, springing, cohesiveness, chewiness, and gumminess. These parameters are essential for evaluating sensory qualities such as tenderness and firmness. An increase in muscle hardness correlates with greater resistance to deformation and typically higher springing. Cohesiveness describes the muscle’s ability to maintain structural integrity through internal bonding, while chewiness provides a comprehensive measure of texture. Gumminess quantifies the flow characteristics of semi-solid products under applied force [36,37]. In this trial, elevated threonine levels significantly enhanced the cohesiveness, elasticity, gumminess, chewiness, and adhesiveness of rice field eel muscle, with the most pronounced improvements observed in the T4 and T5 groups. These results align with findings in red swamp crawfish (*Procambarus clarkii*) [38], suggesting that threonine may optimize dietary amino acid balance, facilitate protein synthesis, and consequently modify muscle texture characteristics to enhance palatability. However, the hardness of rice field eel muscle exhibited an initial decline, followed by an increase as threonine levels rose. This nonlinear response implies that textural properties may be influenced by extrinsic factors, including feed composition, aquaculture practices, and slaughter methods. Therefore, continuous refinement of farming techniques could enable the targeted modulation of fish muscle texture to better align with consumer preferences.

Fish muscle growth represents a dynamic process involving two distinct mechanisms: muscle fiber hypertrophy and hyperplasia [39]. Muscle fibers play a critical role in maintaining meat quality characteristics and coloration [40]. Research indicates that in fish and other vertebrates, muscle fiber growth is specifically regulated by several myogenic regulatory factors (MRFs), including MYF5, MyoD1, MRF4, and MyoG [41]. Experimental results demonstrated that the relative expression levels of MyoG, MyoD1, and MYF5 genes initially increased and subsequently decreased as dietary threonine levels rose, reaching their maximum values in the T4 group. In contrast, MRF4 gene expression exhibited a continuous upward trend with increasing threonine levels, achieving peak expression in the T6 group. The findings suggest that dietary threonine enhances the expression of myofiber growth-related genes in rice field eel, resulting in improved muscle quality. This is consistent with previous research on dietary threonine in hybrid catfish [13]. The results indicate that dietary threonine may upregulate MRF gene expression in muscle tissue, thereby stimulating myoblast proliferation and differentiation. This mechanism regulates myofiber development and ultimately influences muscle growth.

### 4.3. Biochemical Analysis of Serum, Liver, Intestinal, and Muscular Parameters

Serum biochemical parameters are widely used in fish diagnostics to assess nutritional metabolism and health status [42]. Serum TP and ALB levels serve as reliable indicators of hepatic synthetic capacity [43]. This study demonstrated that graded increases in dietary threonine significantly elevated serum TP and ALB concentrations, peaking in the T6 group. These findings align with previous studies on lysine supplementation in rice field eel [44]. The results indicate that although serum TP and ALB continue to rise when essential amino acids exceed optimal levels, this elevation is often accompanied by elevated blood ammonia and transaminase activities, suggesting nitrogen overload from amino acid excess. Serum GPT and GOT represent crucial biomarkers for assessing hepatic injury [45]. In rice field eel, serum GOT activity followed a triphasic pattern—rising, falling, and rising again—with the nadir in the T4 group. Conversely, serum GPT activity declined initially and then increased, with T4 values significantly lower than those of T5. Hepatic GOT activity rose and then fell with increasing threonine, whereas hepatic GPT activity mirrored this trend; both activities were highest in the T4 group. These observations mirror those reported for Indian catfish [10] and Jian carp [35]. This is likely attributable to the optimal threonine dose (T4) restoring amino-acid homeostasis, enhancing hepatic protein synthesis, and preserving membrane integrity, thereby minimizing cytoplasmic leakage of GOT and GPT and yielding the lowest serum activities. Conversely, suboptimal (T1–T3) or excessive (T5–T6) doses disrupt this balance, increasing hepatocyte membrane permeability or exacerbating oxidative stress and ultimately causing enzyme activities to rebound.

Under normal physiological conditions, the antioxidant system maintains a dynamic balance between free-radical production and elimination. When this balance is disrupted, excessive free radicals trigger oxidative damage and stress, ultimately impairing growth and muscle quality. Aquatic animals employ a range of antioxidant enzymes to combat oxidative stress, primarily including T-AOC, CAT, and SOD [46,47]. In this study, graded threonine supplementation significantly elevated hepatic CAT and SOD activities, whereas intestinal CAT and T-AOC first increased and then declined. Muscle CAT activity followed the same trend, peaking in T4. These results indicate that optimal threonine enhances antioxidant capacity in both liver and intestine. This is consistent with previous findings in mid-growth grass carp [11] and half-smooth tongue sole (*Cynoglossus semilaevis*) [48]. Moreover, hepatic MDA content declined linearly with increasing threonine, and levels in T2, T4, and T5 were significantly lower than in the control group. Thus, threonine possesses anti-inflammatory and antioxidant properties, and optimal levels protect fish by mitigating oxidative damage. These findings demonstrate that threonine alleviates oxidative stress in the liver, intestine, and muscle, likely by attenuating lipid-peroxidation toxicity and lowering hydrogen peroxide levels, thereby preventing cellular damage [49,50].

### 4.4. Analysis of Intestinal Digestive Enzyme Activity and Histomorphology

Fish growth is intrinsically linked to nutrient absorption and digestion, and digestive-enzyme activity serves as a key physiological indicator [51]. The principal digestive enzymes—protease, lipase, and amylase—catalyze the hydrolysis of macromolecules (proteins, lipids, and starches) into absorbable peptides, amino acids, fatty acids, glycerol, and oligosaccharides. These activities are essential for somatic growth and development. Our findings reveal a dose-dependent response in rice field eels, where intestinal lipase and trypsin activities exhibit a quadratic trend with increasing threonine levels, peaking in the T4 treatment group. Comparable results have been documented in juvenile Jian carp [35] and mid-growth grass carp [11]. These results indicate that optimal dietary threonine enhances intestinal digestive enzyme activity and nutrient absorption, thereby improving growth performance. However, dietary threonine levels showed no significant effect on intestinal amylase activity in rice field fish, differing from observations in pigeons (*Columba livia domestica*) [52], likely reflecting species-specific divergence. Greater villus height increases absorptive surface area, enhances nutrient–epithelium contact, and elevates brush-border enzyme densities, collectively optimizing digestion and absorption. This study reveals that dietary threonine deficiency causes partial shedding and damage to intestinal villi, consistent with findings in terrestrial species where threonine deprivation leads to intestinal structural impairment, microvilli loss, and lamina propria relaxation. Such structural damage directly compromises intestinal function, impairing nutrient digestion and absorption [53]. VH in the T4 group was significantly greater than in controls, indicating that optimal threonine enhances villus elongation, expands absorptive surface area, and improves digestive efficiency. Similar outcomes have been reported in triploid rainbow trout [54]. Collectively, threonine is indispensable for optimizing nutrient digestion and absorption and for maintaining intestinal morphological integrity.

### 4.5. Analysis of Gut Microbiota

Techniques like 16S rRNA gene amplification are effective tools for studying complex microbial communities in animals and plants, with broad applications in gut microbiota research. Gut microbes form symbiotic relationships with their hosts and play critical roles in digestion, nutrient metabolism, growth, and immune function, thereby being essential for aquatic animal health. The composition and structure of these microbial communities significantly correlate with fish growth performance and digestive efficiency [55]. Our findings demonstrate distinct microbial community structures among the T1, T4, and T5 groups. Notably, the T4 group exhibited significantly higher total OTUs, unique OTUs, and Shannon index values compared to T5, suggesting that optimal threonine supplementation enhances microbial diversity while preserving ecosystem stability. Studies on the intestinal microbiota of largemouth bass similarly demonstrated that methionine supplementation enhances both the evenness and richness of gut microbial communities [56]. Dietary threonine levels had no significant effects on ACE or Chao indices, consistent with Jiang Qingqing’s observations [38] that threonine supplementation in feed did not alter the intestinal microbial diversity of red swamp crayfish (*P. clarkii*). The current study identified *Proteobacteria* as the dominant phylum in the intestinal microbiota of rice field eel. This phylum is prevalent in fish intestines, with several species of *Proteobacteria* showing strong functional correlations with host digestion. These microorganisms support intestinal health and promote fish growth by producing polyhydroxybutyrate, an energy source for intestinal epithelial cells [57,58]. At the taxonomic level, the dominant bacterial genera in the rice field eel intestine were primarily *Pseudomonas*, *Acinetobacter*, *Cetobacterium*, and others, which is consistent with previous findings [58].

This study identified six differentially abundant genera categorized into two functional groups: Positive drivers included *Corynebacterium* (*Proteobacteria*) and *Methylocella* (*α-Proteobacteria*), which play critical roles in amino acid metabolism by supplying energy and micronutrients to the host and generating ATP through short-chain organic acid utilization, respectively [59,60]. Negative regulators included *Clostridium_sensu_stricto_13* (*Firmicutes*), which is known to impair protein efficiency [61], and *Brevinema* (*Spirochaetes*), which is associated with intestinal inflammation and epithelial damage—potentially inhibiting fish growth through oxidative stress induced by unknown virulence factors [62], as well as *Sphingobacterium* and *Dietzia*. *Corynebacterium* and *Methylocella* abundance positively correlated with weight gain, muscle texture, and MRF4 expression. PICRUSt2 functional prediction indicated that the T4 group increased the abundances of Corynebacterium and *Methylocella*, activating three key metabolic pathways: glycine/serine/threonine, pyruvate, and glyoxylate, thus improving energy and amino acid availability. These results suggest that dietary threonine supplementation promotes muscle growth by enhancing amino acid metabolism and TCA cycle activity. This mechanism parallels findings in Hu sheep, where *Prevotella* species boosted acetate metabolism, subsequently increasing TCA cycle flux, pyruvate metabolism, and amino acid metabolism to generate more energetic compounds that stimulated growth [63]. *Clostridium_sensu_stricto_13* and *Brevinema* were negatively correlated with protein efficiency, as well as with MyoD1 and MRF4 expression. PICRUSt2 functional annotation indicated that these bacterial genera promote intestinal inflammation, impair carbon and nitrogen substrate utilization, and downregulate MyoD1, MyoG, and MRF4 expression, ultimately reducing PER and compromising muscle quality in rice field eels while inhibiting muscle growth. Similar observations in Jinbei carp [64] suggest that *Clostridium*/*Spirochaeta* may commonly inhibit myogenesis in fish through the ammonia–inflammation–protein degradation axis.

## 5. Conclusions

The addition of threonine to the feed improved amino acid balance, upregulated the expression of myogenic regulatory genes, and reshaped the intestinal microbiota, thereby significantly enhancing the growth performance and muscle quality of rice field eel. Dietary supplementation with 0.9% threonine yielded optimal growth performance and enhanced muscle characteristics, including increased crude protein content, decreased crude fat content, and improved cohesiveness, springing, and chewiness. Additionally, KEGG functional prediction indicated that 0.9% threonine activated the glycine/serine/threonine–pyruvate–glyoxylate metabolic pathway by increasing the relative abundance of *Corynebacterium* and *Methylocella*, thereby providing energy and amino acid substrates for muscle fiber hyperplasia. Simultaneously, it inhibited the ammonia–inflammation–muscle protein degradation axis mediated by *Clostridium_sensu_stricto_13* and *Brevinema*. The addition of 0.9% threonine enhanced lipase and trypsin activities, significantly elevated liver CAT and SOD activities, and reduced MDA content, indicating simultaneous improvements in antioxidant and digestive capacities. Based on broken-line regression analysis, the optimal threonine requirement for rice field eel was determined to be 0.75–0.87%. Considering the comprehensive research findings, a threonine supplementation of 0.77–0.9% (7.5 g/kg–9 g/kg) can serve as an effective and sustainable nutritional additive for rice field eel, laying a foundation for promoting the sustainable and healthy aquaculture of this species. Future studies should further elucidate the specific synergistic mechanisms of the gut–muscle axis at this dosage to evaluate its broader potential for enhancing muscle quality.

## Figures and Tables

**Figure 1 animals-15-02643-f001:**
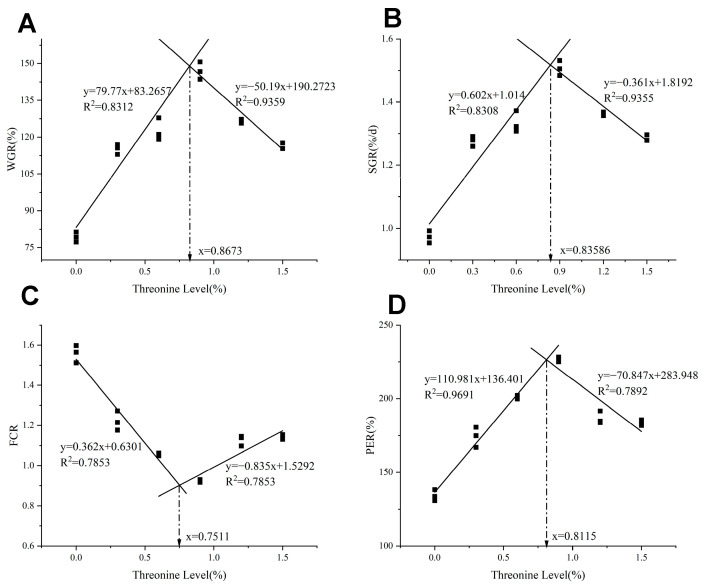
The relationship between dietary threonine levels and (**A**) WGR: weight gain ratio; (**B**) SGR: specific growth rate; (**C**) FCR: feed conversion ratio; and (**D**) PER: protein efficiency ratio in rice field eel. The figure was drawn based on the theoretical addition amount of threonine, and the obtained optimal addition amount is the theoretical value.

**Figure 2 animals-15-02643-f002:**
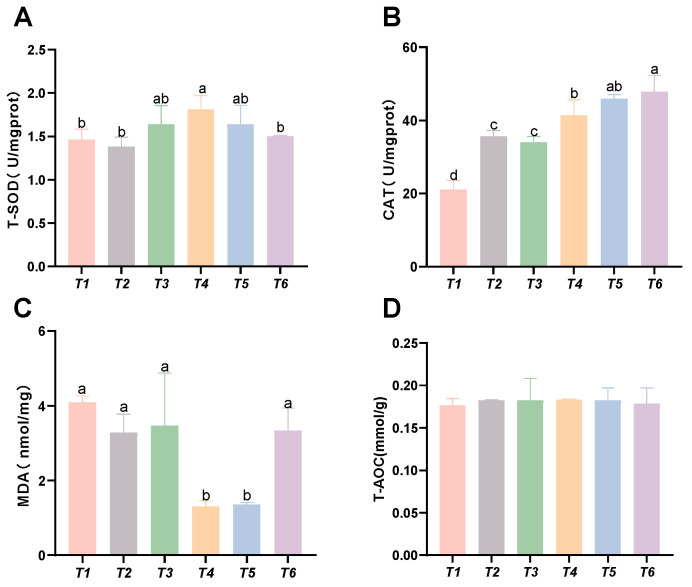
Effects of dietary threonine levels on antioxidant enzyme activities in the liver of rice field eel. (**A**) T-SOD: total superoxide dismutase; (**B**) CAT: catalase; (**C**) MDA: malondialdehyde; and (**D**) T-AOC: total antioxidant capacity; values with no letter or the same letter superscripts are not significantly different (*p* > 0.05), while different small letter superscripts indicate a significant difference (*p* < 0.05).

**Figure 3 animals-15-02643-f003:**
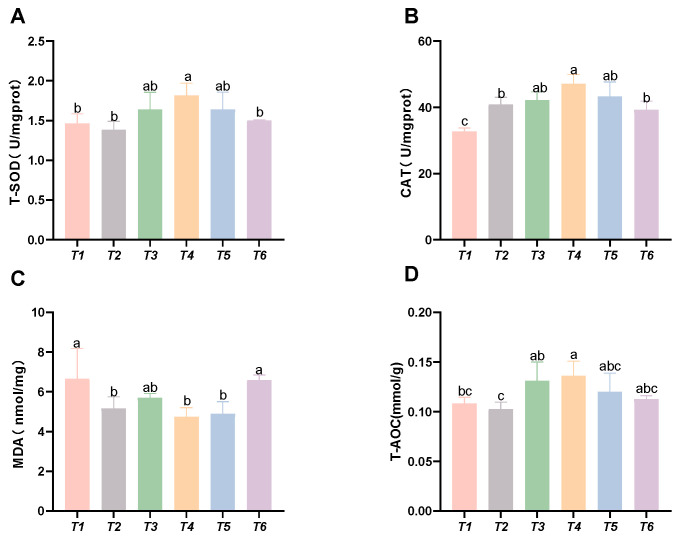
Effects of dietary threonine levels on antioxidant enzyme activities in intestine in rice field eel. (**A**) T-SOD: total superoxide dismutase; (**B**) CAT: catalase; (**C**) MDA: malondialdehyde; and (**D**) T-AOC: total antioxidant capacity; values with no letter or the same letter superscripts are not significantly different (*p* > 0.05), while different small letter superscripts indicate a significant difference (*p* < 0.05).

**Figure 4 animals-15-02643-f004:**
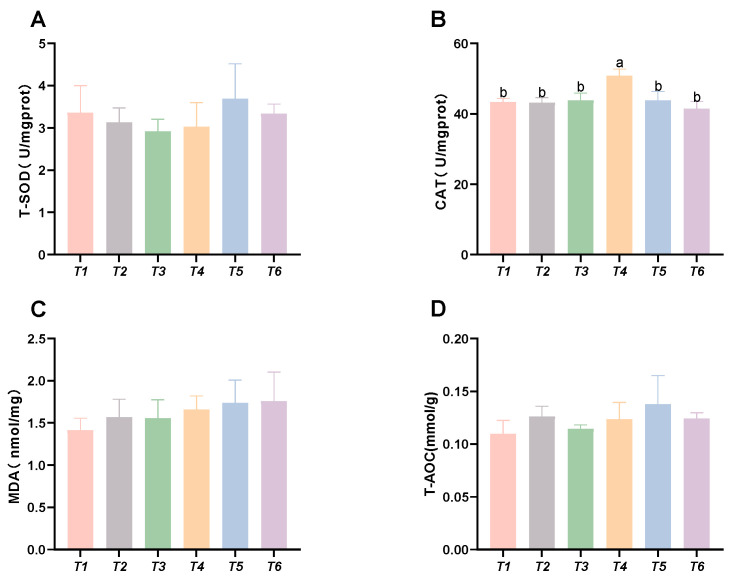
Effects of dietary threonine levels on antioxidant enzyme activities in muscle in rice field eel. (**A**) T-SOD: total superoxide dismutase; (**B**) CAT: catalase; (**C**) MDA: malondialdehyde; and (**D**) T-AOC: total antioxidant capacity; vales with no letter or the same letter superscripts are not significantly different (*p* > 0.05), while different small letter superscripts indicate a significant difference (*p* < 0.05).

**Figure 5 animals-15-02643-f005:**
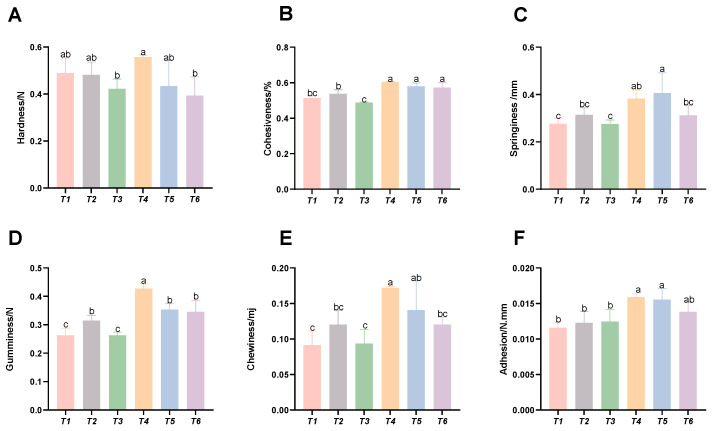
Effects of dietary threonine levels on muscle texture in rice field eel. (**A**) Hardness; (**B**) cohesiveness; (**C**) springiness; (**D**) gumminess; (**E**) chewiness; and (**F**) adhesion; values with no letter or the same letter superscripts are not significantly different (*p* > 0.05), while different small letter superscripts indicate a significant difference (*p* < 0.05).

**Figure 6 animals-15-02643-f006:**
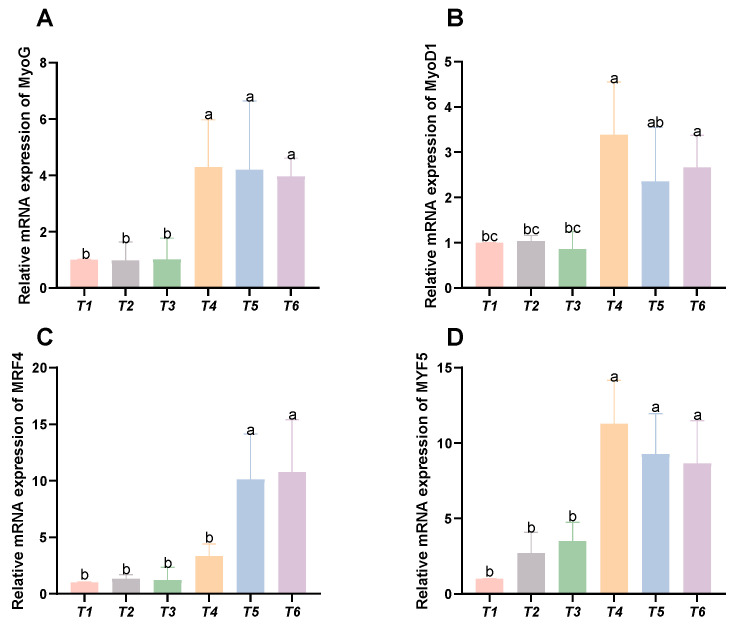
Effects of dietary threonine levels on the expression levels of genes related to myofiber growth in rice field eel. (**A**) MyoG; (**B**) MyoD1; (**C**) MRF4; and (**D**) MYF5; values with no letter or the same letter superscripts are not significantly different (*p* > 0.05), while different small letter superscripts indicate a significant difference (*p* < 0.05).

**Figure 7 animals-15-02643-f007:**
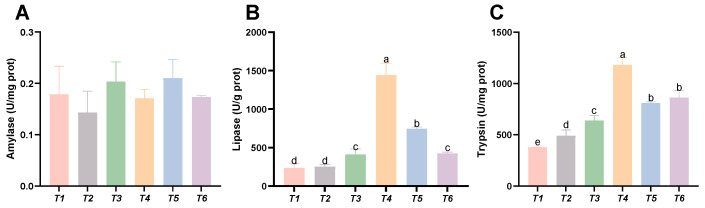
Effects of dietary threonine levels on digestive enzyme activities in intestine in rice field eel. (**A**) Amylase; (**B**) lipase; and (**C**) trypsin; values with no letter or the same letter superscripts are not significantly different (*p* > 0.05), while different small letter superscripts indicate a significant difference (*p* < 0.05).

**Figure 8 animals-15-02643-f008:**
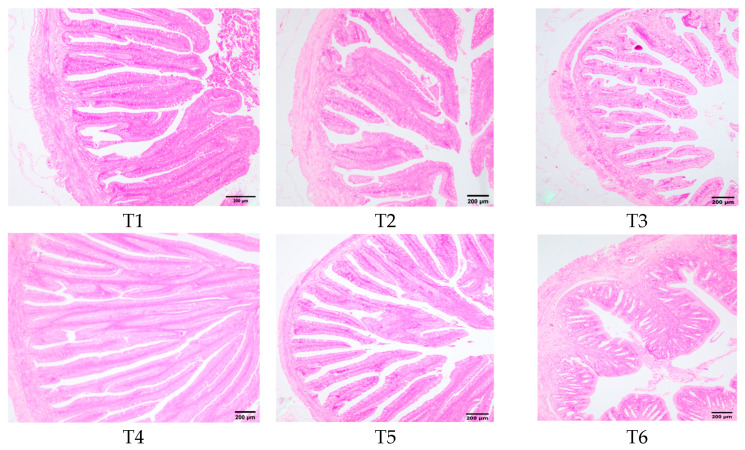
Effects of dietary threonine levels on the morphological structure in rice field eel intestinal tissue (100×).

**Figure 9 animals-15-02643-f009:**
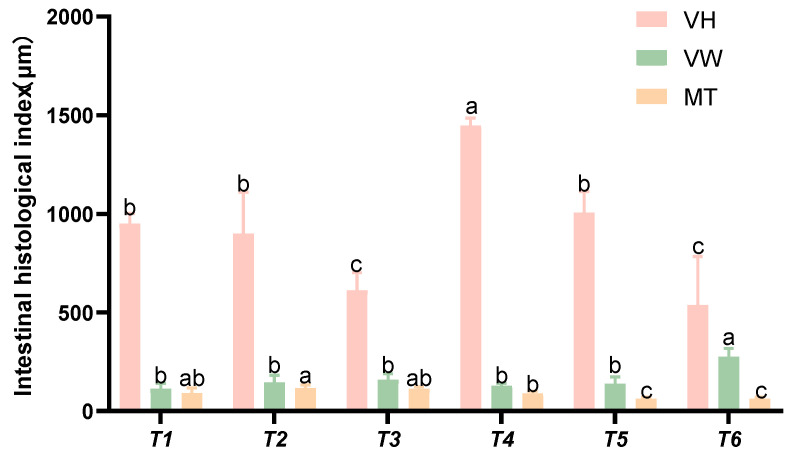
Effects of dietary threonine levels on the morphological indices of the intestinal tissue in rice field eel, with different letters indicating significant differences among different groups (*p* < 0.05).

**Figure 10 animals-15-02643-f010:**
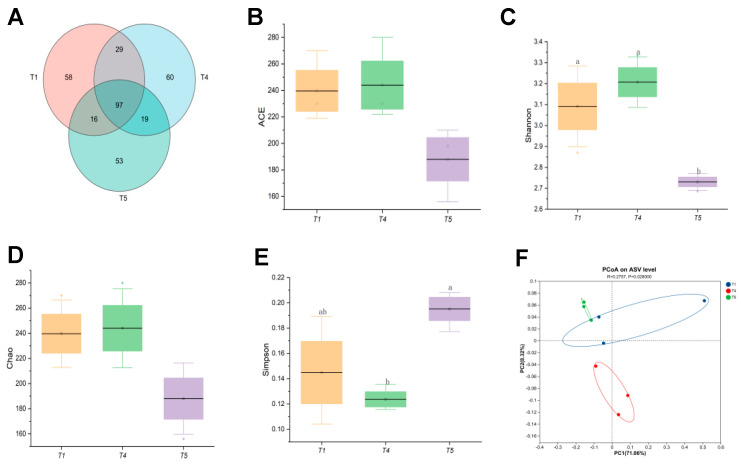
Effects of dietary threonine levels on intestinal microbial diversity in rice field eel. (**A**) Venn; (**B**) ACE; (**C**) Shannon; (**D**) Chao; (**E**) Simpson; (**F**) PCoA; with different letters indicating significant differences among different groups (*p* < 0.05).

**Figure 11 animals-15-02643-f011:**
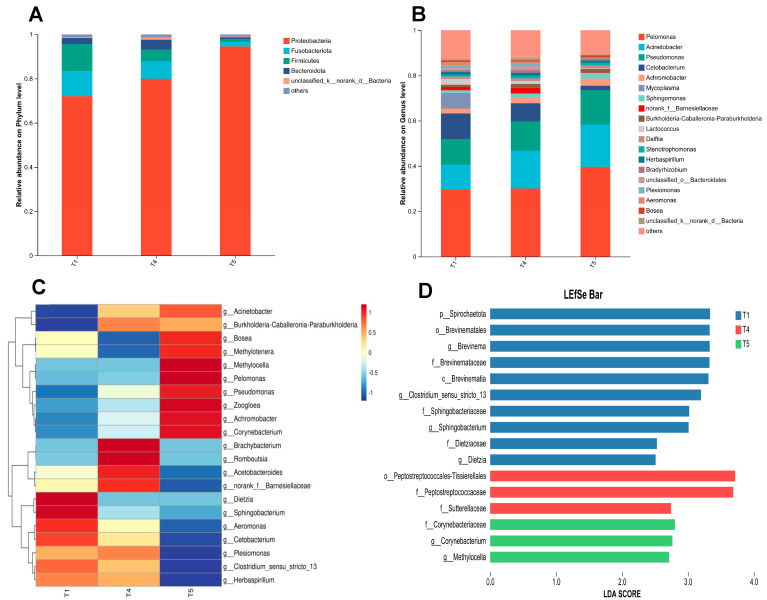
Effects of dietary threonine levels on the composition and variation of intestinal microbiota in rice field eel. (**A**) Relative abundance on Phylum level; (**B**) Relative abundance on Genus level; (**C**) Community heatmap; (**D**) Lefse analysis.

**Figure 12 animals-15-02643-f012:**
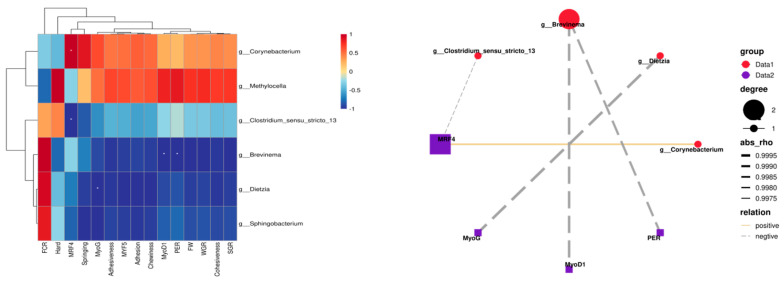
Correlation analysis between differential bacterial genera and growth performance, muscle quality, and growth parameters, with the same row are marked with one *, indicating a significant difference (*p* < 0.05).

**Figure 13 animals-15-02643-f013:**
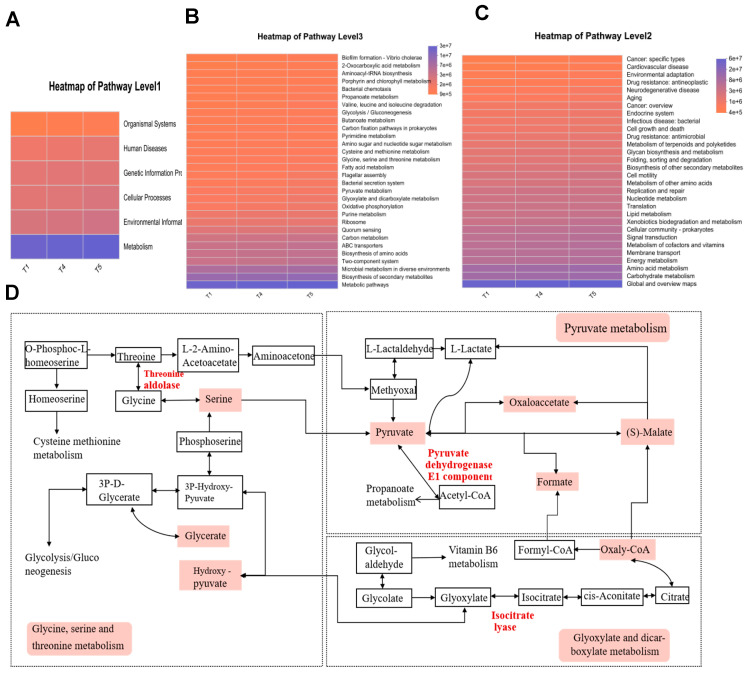
Functional prediction of gut microbiota based on KEGG metabolic pathway analysis. (**A**) Heatmap of Pathway 1; (**B**) Heatmap of Pathway 2; (**C**) Heatmap of Pathway 3; (**D**) KEGG Pathway.

**Table 1 animals-15-02643-t001:** Diet composition and nutritional level (%).

Ingredient	T1	T2	T3	T4	T5	T6
Fish meal	15	15	15	15	15	15
Soy protein concentrate	5	5	5	5	5	5
Peanut meal	30	30	30	30	30	30
Corn protein powder	25	25	25	25	25	25
Fish oil	3.6	3.6	3.6	3.6	3.6	3.6
Soybean lecithin	2	2	2	2	2	2
Wheat meal	8.1	8.1	8.1	8.1	8.1	8.1
Amino acid mix ^1^	7.3	7.3	7.3	7.3	7.3	7.3
Choline chloride	0.5	0.5	0.5	0.5	0.5	0.5
Ca(H_2_PO_4_)	1	1	1	1	1	1
Mineral premix ^2^	0.5	0.5	0.5	0.5	0.5	0.5
Vitamin premix ^3^	0.5	0.5	0.5	0.5	0.5	0.5
Threonine	0	0.3	0.6	0.9	1.2	1.5
Glycine	1.5	1.2	0.9	0.6	0.3	0
Total	100	100	100	100	100	100
Nutrient levels						
Crude protein	47.86	47.08	47.12	47.79	47.53	47.64
Ash	10.59	9.33	9.13	10.37	10.40	10.46
Crude lipid	5.69	5.34	5.36	5.39	5.43	5.50

Note: The addition amount of threonine in the table is the theoretical value. ^1^ Amino acid mixture: arginine, 1.056%; histidine, 0.3035%; isoleucine, 0.986%; leucine, 0.901%; lysine, 1.513%; cysteine, 0.5285%; phenylalanine, 0.3765%; tyrosine, 0.243%; valine, 0.838%; and tryptophan, 0.5%. ^2^ Vitamin premix (per/kg): VA, 4500 IU; VD3, 1000 IU; VE, 200 mg; VK, 40 mg; VC, 25 mg; VB1, 28 mg; VB2, 80 mg; VB6, 40 mg; VB12, 0.01 mg; nicotinic acid, 200.0 mg; pantothenic acid, 80 mg; biotin, 0.2 mg; folic acid, 4 mg; inositol, 80 mg; and choline, 500 mg. ^3^ Mineral premix (per/kg): NaCl, 1000 mg; CuSO_4_⋅5H_2_O, 3.9 mg; FeSO_4_⋅7H_2_O, 180 mg; ZnSO_4_⋅7H_2_O, 70 mg; MnSO_4_⋅H_2_O, 28 mg; MgSO_4_⋅7H_2_O, 50 mg; CoCl_2_, 0.89 mg; and KI, 0.8 mg.

**Table 2 animals-15-02643-t002:** The primers sequences used for quantitative RT-PCR.

Gene Name	Primer Sequence (5′–3′)	Product Length (bp)
*RPL17*	F: GTTGTAGCGACGGAAAGGGAC R: GACTAAATCATGCAAGTCGAGGG	160
*M* *yoG*	F: CGGCAACATTGAGGGAGA R: CTGCTGGTTCAGGGAGGA	173
*M* *yoD1*	F: GGGATTGAGCATGGAGTTG R: GAGGAGGCGGTTGAAGA	174
*MRF4*	F: GCTGGACGAACAGGAGAA R: GCAAGAGGCTGGAGGATG	194
*MYF5*	F: CCGCAATGCCATTCAGTA R: TTATCGTCCAAACCCTCGT	219

**Table 3 animals-15-02643-t003:** Effects of dietary threonine levels on growth performance in rice field eel.

Index	T1	T2	T3	T4	T5	T6
IW/g	18.47 ± 0.65	18.57 ± 0.08	18.48 + 0.19	18.42 ± 0.039	18.40 ± 0.18	18.49 ± 0.10
FW/g	33.11 ± 0.46 ^d^	39.93 ± 0.39 ^c^	41.15 ± 0.99 ^b^	45.50 ± 0.74 ^a^	41.63 ± 0.24 ^b^	40.10 ± 0.09 ^c^
FCR	1.56 ± 0.04 ^a^	1.22 ± 0.05 ^b^	1.06 ± 0.01 ^d^	0.92 ± 0.01 ^e^	1.13 ± 0.03 ^c^	1.14 ± 0.01 ^c^
WGR/%	79.28 ± 2.08 ^d^	115.17 ± 2.05 ^c^	122.71 ± 4.54 ^b^	146.98 ± 3.53 ^a^	126.29 ± 0.83 ^b^	116.86 ± 1.32 ^c^
SGR/%·d^−1^	0.97 ± 0.02 ^d^	1.28 ± 0.02 ^c^	1.33 ± 0.34 ^b^	1.51 ± 0.02 ^a^	1.36 ± 0.01 ^b^	1.29 ± 0.01 ^c^
PER/%	134.19 ± 3.75 ^e^	174.12 ± 6.83 ^d^	200.78 ± 1.26 ^b^	226.30 ± 1.70 ^a^	186.70 ± 4.33 ^c^	183.79 ± 1.90 ^c^

Note: In the same row, values with no letter or the same letter superscripts are not significantly different (*p* > 0.05), while different small letter superscripts indicate a significant difference (*p* < 0.05).

**Table 4 animals-15-02643-t004:** Effects of dietary threonine levels on body index in rice field eel.

Index	T1	T2	T3	T4	T5	T6
CF (g/cm^3^)	0.09 ± 0.01	0.09 ± 0.01	0.08 ± 0.01	0.10 ± 0.05	0.09 ± 0.01	0.09 ± 0.01
VSI (%)	4.87 ± 0.63 ^a^	4.07 ± 0.47 ^b^	3.56 ± 0.68 ^b^	3.71 ± 1.15 ^b^	3.94 ± 0.89 ^b^	3.75 ± 1.16 ^b^
HSI (%)	8.45 ± 0.81 ^a^	7.82 ± 0.85 ^ab^	7.36 ± 1.09 ^b^	7.41 ± 1.33 ^b^	7.56 ± 1.05 ^ab^	7.35 ± 2.02 ^b^

Note: In the same row, values with no letter or the same letter superscripts are not significantly different (*p* > 0.05), while different small letter superscripts indicate a significant difference (*p* < 0.05).

**Table 5 animals-15-02643-t005:** Effects of dietary threonine levels on nutrient composition of the whole body in rice field eel.

Index	T1	T2	T3	T4	T5	T6
Moisture (%)	67.61 ± 1.54 ^b^	68.76 ± 0.12 ^b^	68.66 ± 4.00 ^b^	65.92 ± 2.11 ^b^	65.50 ± 2.15 ^b^	72.87 ± 0.27 ^a^
Crude protein (%)	52.02 ± 0.18 ^c^	48.06 ± 0.57 ^d^	53.99 ± 0.32 ^b^	55.53 ± 0.87 ^a^	50.94 ± 0.47 ^c^	55.11 ± 0.34 ^ab^
Crude lipid (%)	43.13 ± 1.09 ^b^	50.95 ± 1.97 ^a^	36.86 ± 7.19 ^cd^	33.03 ± 0.73 ^d^	42.60 ± 3.17 ^bc^	31.30 ± 0.24 ^d^
Ash (%)	8.15 ± 0.59 ^bc^	6.85 ± 0.23 ^d^	7.69 ± 0.17 ^cd^	8.71 ± 0.09 ^b^	7.14 ± 0.04 ^d^	11.13 ± 1.02 ^a^

Note: In the same row, values with no letter or the same letter superscripts are not significantly different (*p* > 0.05), while different small letter superscripts indicate a significant difference (*p* < 0.05).

**Table 6 animals-15-02643-t006:** Effects of dietary threonine on serum biochemical indexes in rice field eel.

Index	T1	T2	T3	T4	T5	T6
Serum						
TP (g/L)	12.76 ± 5.8 ^b^	13.25 ± 4.29 ^b^	17.7 ± 3.13 ^ab^	21.69 ± 10.63 ^ab^	22.40 ± 10.23 ^ab^	31.80 ± 7.48 ^a^
GOT (U/L)	25.25 ± 1.06 ^ab^	28.5 ± 0.71 ^ab^	23.5 ± 2.12 ^ab^	22.5 ± 6.36 ^b^	28.00 ± 1.41 ^ab^	32.5 ± 4.95 ^a^
GPT (U/L)	5.00 ± 1.73 ^ab^	4.00 ± 1.00 ^ab^	3.33 ± 1.53 ^ab^	3.00 ± 1.41 ^b^	6.33 ± 1.53 ^a^	5.33 ± 1.53 ^ab^
ALB (g/L)	7.57 ± 2.66 ^ab^	6.61 ± 0.58 ^ab^	8.94 ± 2.6 ^ab^	5.51 ± 1.37 ^b^	5.63 ± 1.5 ^b^	10.33 ± 1.8 ^a^
Liver						
GOT (U/L)	33.89 ± 3.97 ^b^	35.72 ± 3.69 ^b^	38.17 ± 0.85 ^b^	43.47 ± 1.46 ^a^	34.75 ± 3.82 ^b^	33.44 ± 2.32 ^b^
GPT (U/L)	35.17 ± 3.24 ^c^	36.64 ± 1.02 ^c^	53.40 ± 1.53 ^b^	57.88 ± 9.01 ^b^	42.12 ± 4.74 ^c^	66.70 ± 2.54 ^a^

Note: In the same row, values with no letter or the same letter superscripts are not significant difference (*p* > 0.05), while different small letter superscripts indicate a significant difference (*p* < 0.05).

## Data Availability

The data that support the findings of this study are available from the corresponding author upon reasonable request.

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
