# Peer review of "Effects of Dietary Threonine Levels on Growth Performance, Biochemical Parameters, Muscle Quality, and Intestinal Microflora of Rice Field Eel (*Monopterus albus*)"

_animals, 2025, doi:10.3390/ani15182643_

Round 1
Reviewer 1 Report
Comments and Suggestions for Authors
The manuscript investigates the impact of dietary threonine levels on growth performance, muscle quality, digestive function, and gut microbiota in rice field eel. The study is well-structured, and the findings are relevant to aquaculture nutrition and sustainable fish production. The results demonstrating optimal effects at 9 g/kg threonine supplementation are novel and provide practical value for formulating cost-effective and eco-friendly diets.
However, these points are suggested for improvement of the manuscript:
Please provide a reference for Lines 56–58.
Please provide a reference for Lines 135–137.
Specify the pest control method that was applied to water hyacinth.
P-values should be added in Tables 3, 4, 5, and 6.
Some figures cited in the manuscript were not found. Please ensure that all figures are cited in the text in the correct sequence.
Author Response
We have provided a point-by-point response to the reviewer’s comments. Please see the attachment.

Reviewer 2 Report
Comments and Suggestions for Authors
The present study, "Effects of Dietary Threonine Levels on Growth Performance, Biochemical Parameters, Muscle Quality, and Intestinal Microflora of Rice field eel (Monopterus albus)," showcases outstanding research on the effects of dietary threonine levels on growth performance, muscle quality, digestive function, gut microbiota in rice field eel. The manuscript demonstrates exceptional writing and presentation quality, featuring a clear and logical structure that facilitates effortless navigation. However, to further enhance the manuscript's quality, I have identified several issues incorporated in Pdf that necessitate major revisions.

Author Response

(The authors gave the same response as above.)

Reviewer 3 Report
Comments and Suggestions for Authors
Title: Effects of Dietary Threonine Levels on Growth Performance, 2 Biochemical Parameters, Muscle Quality, and Intestinal Micro-3 flora of Rice field eel (Monopterus albus)
Authors: Denghang Yu, Yujia Liu, Jiaxiang Chen, Jincheng Wan, Jiaqi Zhang and Chi Zhang
The manuscript appears to be well written but there are some questions about the Materials and Methods.
2.1. Experimental Design and Feed Preparation
What physical feature indicates the eel is a male? Is it the size of the fish? Since females tend to become males after spawning how was it confirmed all the fish in the study were males and not females in transition? What was the importance of having all males?
Were each of the three groups (the three replicates) of 30 fish of equal weight at the beginning of the study?
How much threonine was in the base diet? There will be some from the ingredients. This amount should be included in the treatment level.
Do these fish not require additional niacin? There is no B3 in the premix. Is methionine not essential for rice field eels? It is not in the Amino Acid premix. Was it determined that both components were in the feed ingredients at adequate levels?
2.2. Aquaculture Management
Following the four fixed principles, how much feed was given to the fish in each tank, i.e. what feed amounts were used to calculate the feed conversion? Were they fed a certain percent body weight?
What does t and o stand for, t = final and o = initial but what are the words used. Also, t is defined as time (number of days), duration of the study.
2.6. Muscle, Liver and Intestinal Biochemical Indices
This section could be combined with section 2.3. Sample Collection.
Author Response

(The authors gave the same response as above.)

Round 2
Reviewer 2 Report
Comments and Suggestions for Authors
Accept
Author Response
We sincerely appreciate your attention to and support for our research. Your feedback not only enhances the quality of the article but also paves the way for us to further refine the paper. Thanks a lot!
Reviewer 3 Report
Comments and Suggestions for Authors
Because the authors are not including the the amount of threonine found in the base diet (the authors stated they have this data but it will be shared in later research), they need to rethink the requirement levels found when using the the broken-line regression analysis. The X axis could change.
Author Response

(The authors gave the same response as above.)
